# Sleep Deprivation and Insomnia in Adolescence: Implications for Mental Health

**DOI:** 10.3390/brainsci13040569

**Published:** 2023-03-28

**Authors:** Sara Uccella, Ramona Cordani, Federico Salfi, Maurizio Gorgoni, Serena Scarpelli, Angelo Gemignani, Pierre Alexis Geoffroy, Luigi De Gennaro, Laura Palagini, Michele Ferrara, Lino Nobili

**Affiliations:** 1Department of Neurosciences, Rehabilitation, Ophthalmology, Genetics, Maternal and Child Health (DINOGMI), University of Genoa, 16126 Genoa, Italy; 2Child Neuropsychiatry Unit, IRCCS Istituto Giannina Gaslini, 16147 Genova, Italy; 3Department of Biotechnological and Applied Clinical Sciences, University of L’Aquila, 67100 L’Aquila, Italy; 4Department of Psychology, Sapienza University of Rome, 00185 Rome, Italy; 5Body and Action Lab, IRCSS Fondazione Santa Lucia, 00179 Rome, Italy; 6Department of Surgical, Medical, Molecular and Critical Area Pathology, University of Pisa, Azienda Ospedaliera Universitaria Pisana AUOP, 56126 Pisa, Italy; 7Département de Psychiatrie et D’addictologie, AP-HP, GHU Paris Nord, DMU Neurosciences, Hopital Bichat—Claude Bernard, 75018 Paris, France; 8GHU Paris—Psychiatry & Neurosciences, 1 Rue Cabanis, Université de Paris, NeuroDiderot, Inserm, 75019 Paris, France; 9Psychiatric Clinic, Department of Clinical and Experimental Medicine, Azienda Ospedaliero Universitaria Pisana AUOP, 56126 Pisa, Italy; 10Institute of Psychiatry, Department of Neuroscience and Rehabilitation, University of Ferrara, 44121 Ferrara, Italy

**Keywords:** adolescence, mental health, sleep, insomnia, circadian rhythms, psychiatric disorders, mood disorders, COVID-19

## Abstract

Sleep changes significantly throughout the human lifespan. Physiological modifications in sleep regulation, in common with many mammals (especially in the circadian rhythms), predispose adolescents to sleep loss until early adulthood. Adolescents are one-sixth of all human beings and are at high risk for mental diseases (particularly mood disorders) and self-injury. This has been attributed to the incredible number of changes occurring in a limited time window that encompasses rapid biological and psychosocial modifications, which predispose teens to at-risk behaviors. Adolescents’ sleep patterns have been investigated as a biunivocal cause for potential damaging conditions, in which insufficient sleep may be both a cause and a consequence of mental health problems. The recent COVID-19 pandemic in particular has made a detrimental contribution to many adolescents’ mental health and sleep quality. In this review, we aim to summarize the knowledge in the field and to explore implications for adolescents’ (and future adults’) mental and physical health, as well as to outline potential strategies of prevention.

## 1. Introduction

Sleep is a neurophysiological process that plays a key role in biological pathways crucial to brain and body health [1,2,3]. The timing and duration of sleep and wakefulness arise from a complex and dynamic interplay between homeostatic and circadian processes [4]. The homeostatic drive increases with wake duration, indicating the increment in sleep need. The circadian process results from a complex network of organ clocks coordinating external cues with behavior and metabolic outputs. The circadian process favors wakefulness in opposition to the homeostatic pressure to sleep and promotes sleep onset during the nighttime hours. Sleep changes significantly during childhood and adolescence [5], and sleep disorders can be widespread in these age groups.

Adolescence is defined as a time window between 10 and 19 years, straddling childhood and adulthood [6]. This phase of life is marked by the advent of puberty, a unique stage of human development characterized by rapid physical and psychological changes that are followed or preceded by emotional and social upheavals.

Adolescence is characterized by changes in behavior and sleep homeostasis with modifications in the circadian rhythms that may undergo phase delay and a slowing of homeostatic sleep pressure [7]. In this scenario, social interactions and improper use of technology may limit the quantity of sleep with detrimental effects on adolescents’ developmental trajectories. Recently, adolescence has been recognized as the second period, after early childhood, of foundational learning and neural plasticity, with high risk but also great potential for recovering from sick conditions [8].

Adolescent people constitute the largest part of the human population, accounting for one-sixth. Low-income countries are seeing an unprecedented expansion of this age group [6]. Adolescents are now facing a wide array of cultural, technological, and social innovations that are impacting their growth both from a psychological and physical point of view, with an influence also on sleep quality [9] (Figure 1). Indeed, the present context of uncertainty that has followed the COVID-19 outbreak (and its related confinement) [10] and the global context of expanding political violence [11], make it more challenging to interpret future directions and more difficult in general to preserve adolescents’ mental health without any public support.

This review focuses on adolescents’ brain physiology and its implications for sleep changes, which may easily unlock potentially harmful conditions that can undermine or worsen mental health. We also aimed to highlight potential interventions for promoting good health and preventing the detrimental effects of sleep deprivation and social drift.

## 2. Adolescence, Brain Maturation and Mental Health

Adolescence is biologically marked by the advent of puberty, the neurohormonal processes that guarantee sexual maturation, which typically have their onset between the ages of 8–9 years and 13–14 years (depending on whether the subject is of female or male sex) [18,19]. Of interest, puberty begins with specific brain hormone signals that activate a complex neuroendocrine network. The trigger is the pulsatile nocturnal release of the gonadotropin-releasing hormone from specialized hypothalamic neurons, which promotes gonadal growth and allows the acquisition of secondary sexual characteristics commonly named gonadarche. It is usually preceded two years before by the independent maturation of the adrenal gland, with an increase in the production of adrenal androgens (dehydroepiandrosterone and dehydroepiandrosterone sulfate), as well as the expansion of the adrenal zona reticularis; these drive changes in sweat secretion and the emergence of axillary and pubic hair, with poorly understood evolutionary significance. The starter of these complex processes is only partially known, but genetic as well as environmental factors (nutritional, psychological, and socioeconomic conditions, or neoplastic diseases) play a key role. This neurohormonal cascade is accompanied by several body transformations driven by maturation of other hypothalamus-pituitary axes, which include rapid physical growth (supported by the insulin-like growth factor and the thyroid hormones), modifications of metabolic homeostasis, and changes in sleep regulation (in particular, in the circadian rhythms) [20].

Therefore, puberty triggers a complex amalgam of physical modifications that also involve the brain (titratable amounts of sex hormones are measurable at tissue and cerebral circulation levels), affecting individuals’ behavior from a cognitive, emotional, and motivational point of view [21]. These transformations lead to the typical adolescent behavior, commonly seen in other mammals, that covers the tendency towards sensation-seeking and adopting risky behaviors (to promote the removal from the parental nucleus) together with a “natural” circadian phase-delay acquisition, the increase of social interactions with peers (to promote sexual interaction) and increased conflict with parents (to avoid inbreeding).

For instance, modified oxytocin and vasopressin secretion is associated with changes in social interaction and creation of attachments [22].

Regional changes in the human adolescent brain are promoted by different expressions of sexual and adrenal steroid hormone receptors, with a clear role in the development of subcortical regions but also of neocortical maturation. Several studies corroborate the relationship between puberty onset and brain modifications (such as grey matter thinning in favor of greater development of the white matter). Recent research adopted both animal models and neuroimaging techniques to assess regional brain changes (both in gray and white matter) across puberty/adolescence [23]. The most interesting studies addressed the topic of functional connectivity in adolescence, with two major findings: (1) deep brain regions (ventral striatum and amygdala) associated with processing of reward, happiness, and fear increase their neural activity [24]; (2) social brain areas such as the dorsomedial prefrontal cortex, anterior temporal lobe, temporal parietal junction, and superior temporal sulcus develop high specificity [25]. In parallel, the maturation of the prefrontal cortex in toto (including the dorsolateral areas) allows remodeling and the final achievement of more specialized cognitive skills [26].

During the transition from childhood to adulthood, synaptic pruning and formation of selected new synapses occur in the neocortex, with consequent loss of infantile forms of plasticity as well as gain of potentially adolescent-specific synaptic flexibility. A reduction of gray matter volumes as well as an increment of white matter volumes has been observed by structural MRI studies, and these are also associated with reductions of EEG power of up to 40% both in wakefulness and in sleep [27]. The prefrontal cortex rapidly becomes more innervated by amygdala inputs and dopaminergic inputs from the ventral tegmental area (and vice versa), reinforcing the dopaminergic circuits of reward [21]. These functional and anatomical changes in the dopaminergic reward system enable the adolescent brain to engage in physiologically risk-taking behaviors that, when coupled with effective neural regulation, guarantee a positive motivation to adaptation to early adulthood [12]. On the other hand, this temporary hyperfunctioning of the dopamine circuits leads to emotional lability and poor self-regulation, that in at-risk contexts is associated with high rates of depression, anxiety, conduct disorder, and rule-breaking behaviors [12,28]. Recent evidence from animal models and developmental neuroscience studies in human youth has shown that disruption of reward processing enhances the development of internalizing and externalizing disorders during adolescence [29].

Indeed, mental health problems account for 16% of the global burden of disease in people aged 10–19 years and nearly a half of mental issues arise by the age of 14 years, frequently remaining misdiagnosed. From childhood to adolescence, the risk of mood disorders increases by five times and the mortality rate due to treatable causes (such as self-harm or unintentional death/injuries) by three times, with suicide being the third greatest cause of death among people aged 15–19 years [30]. A further issue is that the end of puberty (and maturation of the sexual system) does not coincide with the end of adolescence in high-income countries (and in many developing ones). The adolescence time window has dilated during recent decades due to the delay in young people partaking fully in adult life due to longer periods of economic and individual dependence on the original parental nucleus; this potentially expands the period of major vulnerability [20,21]. The expanded adolescent phase has been considered by many authors to be the result of the industrial revolution that progressively provided improvements in nutritional conditions, lowered the incidence of infectious diseases in childhood and associated mortality, and required an extension of the time spent in formal education.; such changes altered the social perspective about marriage and affective relationships, and in recent decades about the availability of contraception for the youngest. These achievements have modulated the transition to adult life [31,32]. Developmental neuroimaging studies have shown a complex model of brain maturation in adolescence that does not depend only on top-down cortical maturation of the decision-making dorsal prefrontal cortex. Other deep circuits involving the ventral prefrontal cortex and limbic areas have different timings for maturation, which occur at different stages of adolescence, mediated by social and affective processing, through cascades of sexual hormones. The interrelationships existing between the modern social context and the biological changes underpinning the acquisition of adults’ behavior are still poorly understood. Understanding the mechanisms happening in adolescence (a period of social-affective engagement and goal flexibility, but also of high health risk) requires therefore not only biological research but also a social-anthropological perspective in the sense that environment can shape brain modulation through a sort of bottom-up mechanism. This perspective can provide a key for therapeutical approaches [24].

## 3. Sleep Deprivation, Adolescents’ Behavior and Mental Health

Sleep occupies more than a third of adolescents’ daily life and is a pillar supporting a correct physiological restoration of brain and body homeostasis, including higher cognitive functions and emotional regulation [27,33]. Adolescents’ sleep patterns experience physiological modification due to maturation in the reproductive hormonal system, a mechanism which is seen in many other mammals but not completely understood. These changes lead to a global reduction in sleep duration, with a decrease in slow-wave sleep and a slight increase in rapid-eye-movement (REM) sleep [34]. Animal models (corroborated by studies on humans) show that theta oscillations in REM sleep have regional specialization (like brain maturation), with higher synchronization between the hippocampus, the amygdala, and the prefrontal cortex, which has important implication for emotional regulation during sleep. In this sense, the increment in REM sleep during adolescence may have a role in regulating emotional processes [35] and constitutes a strong endophenotype, highly from the environment [36].

Changes in endogenous circadian rhythms are associated with the sleep–wake modifications occurring during adolescence [37]. Of note, profound modifications in endogenous circadian rhythms characterize the transition from childhood to adulthood (Figure 1) [37]. A landmark modification involves melatonin rhythms, which are delayed with puberty [13]. Melatonin is a pineal gland hormone that plays a key role in regulating the sleep-wake cycle, and changes in its daily fluctuation have major effects on sleep patterns [38,39]. The melatonin shift phase is strictly linked with gonadal hormones, and it occurs in conjunction with sexual maturity reached during adolescence [40]. Furthermore, cortisol secretion has a circadian rhythm with high cortisol levels at morning awakening and then decreasing levels until reaching a minimum around bedtime. During adolescence, cortisol levels increase and follow a flatter rhythm [41]. Finally, data on changes in the circadian rhythms of body temperature suggest that the rhythms may be delayed during adolescence, with a delayed timing of the body temperature rhythm associated with eveningness in adolescents [37].

The widespread use of technology at nighttime can further amplify melatonin dysregulation among the younger population [14,42]. Light is a crucial zeitgeber that controls several biological processes, including daily melatonin variations [43]. The short-wavelength-enriched screen light of most modern electronic devices (e.g., smartphones, computers) can mimic the effects of sunlight and disrupt sleep-wake rhythms by dampening nocturnal melatonin secretion [44,45]. Recently, approaches have been implemented to mitigate these effects (e.g., short-wavelength light filters); however, findings reported in the literature do not support such strategies as interventions to improve sleep [46]. In addition to the changes in the circadian timekeeping system, adolescence is also characterized by developmental modifications in the homeostatic sleep regulation process. Studies addressing this topic showed that sleep pressure accumulation is slower in mature adolescents than in prepubertal or early pubertal children [47]. Similar findings were obtained by studies evaluating the effects of extended wakefulness paradigms, confirming lower sleep propensity in mature than in prepubertal adolescents [48]. Finally, social pressure from peers, refusal to follow social/family norms, and increased emancipation in deciding bedtime routines, can further fuel sleep timing modifications in adolescence [7,49].

Overall, the combination of biological, environmental, and psychosocial factors that interact with the critical sleep regulatory systems determines the adolescents’ well-documented tendency towards eveningness [37]. From puberty onwards, chronotype is gradually delayed, reaching its peak rate of change at around 20 years old and then advancing across the lifespan [50,51]. However, the adolescents’ evening-oriented circadian clock is not attuned to the morning-oriented societal demands, leading them to experience the largest discrepancy between the endogenous biological and the exogenous social clock (i.e., social jetlag) [50]. A recent study supported this idea by analyzing data from over 18 thousand people aged 0–25 years [52]. The authors found that the social jetlag (the difference between weekday and weekend sleep schedules) progressively increases until 16–17 years old, when it reaches its maximum peak. A similar but opposite trend was found for the time in bed measure, because weekday get-up times became gradually earlier until 17 years while, on the contrary, bedtimes became later. These findings are consistent with the assumption that early school start times inevitably limit adolescents’ amount of sleep, exposing most of them to a chronic sleep deprivation condition [53,54]. The American Centers for Disease Control and Prevention (CDC) estimated that 72.7% of high school students sleep less than 8 h per night during weekdays [55]. The scientific literature confirms a pervasive sleep deprivation condition among adolescents worldwide, which seems even more significant in Asian countries [56]. The adolescent sleep loss epidemic transversely affects modern societies due to its broad-spectrum public health implications. Findings from the Youth Risk Behavior Survey, conducted by the CDC on a nationally representative sample of 67 thousand high schoolers, suggested an alarming dose-dependent relationship between school-night sleep duration and unsafe behaviors [57]. Sleeping 7, 6, or less than 6 h per night was associated with gradually increased odds of risky driving and sexual activities, aggressive behaviors, and use of tobacco, alcohol, marijuana, and other drugs. Moreover, another analysis of Youth Risk Behavior Survey data revealed that adolescents sleeping less than 7 h per school night are more prone to several injury-related risk behaviors than those sleeping 9 h [58]. Studies on large data sets from high school students corroborate these findings [15,59].

As mentioned in the previous section, the differential maturation timing of brain structures and the consequent imbalance between cognitive and affective control systems helps to explain the increased tendency to risky behaviors in adolescence. Sleep problems have been proposed to exacerbate this scenario, as studies using functional magnetic imaging (fMRI) scans suggested that risky behaviors induced by poor sleep are associated with an unbalanced activation of cognitive (prefrontal cortex) and affective control systems (insula and ventral striatum) [60]. Furthermore, adolescence represents the lifetime period in which most psychiatric disorders emerge [61], and mounting evidence has supported the crucial role of inadequate sleep time in developing psychiatric conditions. Sleep deprivation could lead to aberrant myelination of the uncinate fasciculus which is the preferential pathway of connection between the limbic system, the orbitofrontal and medial prefrontal cortices, therefore generating potential aberrant emotional regulatory pathways that can lead to lower functioning of the so-called “social brain” and therefore to potential psychiatric conditions [38,62]. In this sense, in addition to sleep deprivation, alterations in sleep macrostructure, such as alterations in REM sleep (which is known to have role in the maturation of these circuits) [35,36] may have a role in enhancing this vicious cycle. A study has highlighted also that fragmentation of REM sleep and the increase of its density in adolescents is associated with mood disorders, but further evidence is needed [63]. Cross-sectional investigations on large adolescent samples systematically showed an association between sleep loss and mental health [64,65,66]. This evidence has been corroborated by prospective studies that confirmed a causal link between inadequate sleep and the subsequent onset of mental health disorders, such as anxiety and depression. Analyzing longitudinal data referring to 3 thousand US individuals aged 11–17, Roberts and Dong demonstrated that sleeping less than 6 h per school night substantially increased the risk for future anxiety and depression symptoms one year later [67,68].Recent findings have shown that alterations in sleep and circadian rhythms are also predictive factors of non-suicidal self-injuries (NSSIs), suicidal ideation (SI), and suicidal attempts (SA) [69]. These behaviors are particularly prevalent among adolescents, as NSSIs seem to affect 18% of adolescents worldwide [70]; 14% of them report SI [71], while 8.6% have attempted suicide [72]. Investigations from the US to China have shown a significant association between short sleep duration and an increased risk of NSSIs when analyzing large adolescent datasets [57,73]. NSSI behavior is one of the strongest antecedents of suicidality [74], and prospective studies identified insufficient sleep as an essential predictor of transition from NSSIs to SA in a 16-years old sample [75]. Finally, a direct relationship between short sleep duration and suicidality in adolescents has been consistently reported. A systematic review and meta-analysis summarized the current results in this field [76], highlighting a curvilinear relationship between sleep amount and the risks of SI and SA, with the lowest risk levels at 8 h and 8–9 h of sleep per day, respectively. A linear dose-response association between sleep duration and suicide plans was also obtained, indicating that the suicide plan risk increases by 11% for every 1-h decrease in sleep time.

In summary, the current literature is consistent in describing a societal sleep debt among adolescents. The pervasive sleep deficit is driven by the misalignment between the psychophysiological delayed chronotype and the enforced social schedules (i.e., the early school start times) [27,38,62], which surely has an impact on adolescents’ cognitive performance [77]. This scenario gives rise to global public health and developmental concerns due to the broad implications for risky/unsafe behaviors and mental health problems across adolescence and early adulthood.

## 4. Insomnia in Adolescence

Circadian rhythm disturbances and sleep deprivation are frequent in adolescents, as described previously [78]. To note, these disorders may contribute to the onset of insomnia. According to the International Classification of Sleep Disorders Third Edition (ICSD-3) and the Diagnostic and Statistical Manual of Mental Disorders Fifth Edition (DSM-5) [79,80], insomnia is described as a significant concern with sleep, which includes difficulty in sleep initiation, sleep maintenance, and/or morning awakening in the context of adequate sleep opportunity. The diagnosis is made if the sleep difficulties are responsible for clinically significant distress or impairment in social, occupational, educational, academic, behavioral, or other important areas of functioning. Daytime symptoms include daytime sleepiness, fatigue, decreased performance and cognitive function, low mood, behavioral problems such as hyperactivity, impulsivity and aggression, and interpersonal problems [79,80].

Insomnia has been generally estimated to be a problem in about 20–25% of young people, but its prevalence was surely underestimated due to the concomitant inclusion of the pediatric population along with adolescents [81]. To note, data from the literature indicate that insomnia during adolescence is underreported, under-diagnosed (or mis-diagnosed) and, therefore, under (or mis-) treated [15].

Diagnosing insomnia during adolescence presents unique challenges. First, a diagnosis of insomnia requires the presence of adequate sleep, which is not easily achieved in adolescence [82]. Additionally, delayed sleep phase disorder, which is prevalent in this population, should be excluded from the diagnosis [15].

Insomnia goes far beyond dissatisfaction in sleeping. There is growing evidence of a strong association of insomnia (especially among adolescents) with biologically devastating alterations that include brain cortical misfunctioning, systemic inflammation, and metabolic changes. This situation has caused experts in the field to speak of “insomnia disorder” to cover an umbrella of complex symptomatology that can be highly detrimental in adolescence [83]. Moreover, taken together, data from the literature indicate that insomnia is a common comorbidity in adolescents suffering from mental issues [84,85,86], playing a major role in maintaining and/or exacerbating psychological malaise.

Studies investigating the natural history of chronic insomnia from childhood to adolescence highlight high rates of persistence of this condition, with low full remission in adolescence. Girls, racial/ethnic minorities (including gender minorities), children of low socioeconomic status, those already suffering from psychological, neurological or metabolic disorders, and subjects with an evening chronotype should be monitored for chronicity or development of insomnia symptoms [87,88].

## 5. Insomnia as a Risk Factor for Psychiatric Disorders

Mental disorders have a prevalence of 20% among adolescents, with major depression being the most common, and typically more frequent (with double rate) in females [89]. Adolescent onset of mental disorders is often misdiagnosed and has been related to a chronic behavioral course with higher rates of substance misuse, interpersonal difficulties, and a higher risk of suicide. Identifying modifiable risk factors among adolescents for major mental disorders is thus a priority. Current models looking for modifiable risk factors for the onset of major mental disorders in adolescence have started to include insomnia. In fact, insomnia in the general teenage population is associated with mental health difficulties later in life and increased risk of interpersonal problems and psychiatric disturbances, which may include mood and anxiety disorders, risk-taking behaviors, substance use disorders, and higher suicidal risk [15,16,17]. Although relations between insomnia and psychiatric disorders are bidirectional, the strongest pathways have been shown to extend from early sleep disturbances in youth to later mental disorders [90], highlighting the importance of adolescence as a critical developmental window for preventive strategies. Indeed, insomnia during adolescence is often combined with other sleep disorders, such as insufficient sleep and circadian rhythm disturbances, and the combination of these disorders may confer an even higher risk of developing mental disorders during this particular life period [17].

### 5.1. Mood Disorders

Data have shown that insomnia may confer the risk of developing depression and bipolar disorder in both boys and girls during adolescence [15,16]. In their longitudinal study conducted on 4494 adolescents, Roane and Taylor found that insomnia was a predictor of later depression in adolescents, increasing the risk by 2.3 times [91]. Sivertsen et al. found an increased probability of developing depression in adolescents with short sleep duration (<6 h) in combination with insomnia [92]. A recent meta-analysis showed that insomnia symptoms might confer the risk of developing depression during adolescence with an OR of 1.68 [17]. Insomnia in adolescents may also increase the risk of subsequent bipolar disorder [17,93,94]. Ritter et al. conducted a study in 3021 adolescents and young adults, adjusting analyses for age, gender, parental mood disorder, and lifetime alcohol or cannabis dependence [94]. Poor sleep quality, especially trouble falling asleep and early morning awakening, significantly increased the risk for the subsequent development of bipolar disorder (OR = 1.75; *p* = 0.001). These data were also confirmed in a recent meta-analysis showing that, combined with other sleep disorders such as insufficient sleep and circadian rhythm disturbances, insomnia may confer an even higher risk of developing mood disorders including bipolar disorder, with an OR of 2.68 [17]. In young people who are currently depressed or affected by bipolar disorder, disturbed sleep including insomnia has been related to higher mood severity and recurrence and suicidal ideation [15,16]. Several hypotheses have been reported on this topic and point to the role of biological, psychological, and social mechanisms. Some authors have suggested the possibility that sleep disruption during neurodevelopment may interfere with neuronal plasticity, and with connectivity of the developing brain, thus affecting the development of areas involved in mood and emotion regulation and favoring mood disorders [95]. It has also been hypothesized that insomnia in early adolescence may lead to the development of later depressive symptoms by altering corticolimbic circuitry, possibly via the disruptive effect of hyperarousal and insufficient sleep [96]. Disruption of the corticolimbic circuitry may be a consequence of insomnia, and may impair affective reactivity and regulation [96,97]. Researchers have suggested that disrupted sleep increases vulnerability to anxiety and depression via emotional processes [98]. Insomnia may have a detrimental effect on reward brain circuits during adolescence, leading to mood disorders (in particular, major depression) Authors suggest that insomnia symptoms in early adolescence may disrupt neural reward processing which may be associated with later adolescent depression [96]. In addition, it is possible that sleep disruption in adolescents, by increasing wakefulness in bed, may reinforce ruminative thinking styles, which could develop into hyperarousal and depression over time [90]. Disruptions of circadian rhythms have also been postulated to contribute to the pathophysiology of mood disorders during adolescence and may favor insomnia development as well [13].

### 5.2. Anxiety Disorders

A growing body of research has explored the relationship between sleep problems and anxiety disorders in youth, with evidence for a reciprocal relationship [99]. In particular, the most common problem in anxious youths is the difficulty in initiating or maintaining sleep and this predicts escalation of anxiety symptoms [100]. Some studies have observed an association between sleep and General Anxiety Disorder and Separation Anxiety Disorder [100]. Several hypotheses have been reported on this topic and hypothesized the role of biological and psychological mechanisms. Children and adolescents with anxiety disorders exhibit exaggerated hypoconnectivity between the PFC and amygdala. Hence, it is possible that insomnia and sleep disturbances during neurodevelopment are likely to further compromise emotional regulation by compromising the prefrontal cortex-amygdala functional connections [97]. Alterations of the arousal system may partially explain the close association between insomnia and anxiety in adolescence. The hyperarousal in insomnia may affect the regulation of sleep, arousal, and anxiety which have regulatory overlap in the physiological, neuroanatomical and developmental domains [97]. There is also accumulating evidence that insomnia-related maladaptive cognitions may contribute to a sort of “hypofrontality” and to the relationship between insomnia and anxiety in adolescents [97].

### 5.3. Cognitive Processing, Risk-Taking Behavior, and Substance Use

Insomnia disorders have a negative impact on neurodevelopment, with consequent detrimental effects on cognition, including on high brain skills such as executive functions (among all, first of all, working memory) and on academic achievements, reflecting disruption of brain circuits that typically develop during adolescence (such as dorsolateral-prefrontal cortex and hippocampal/limbic networks) [15,101]. Insomnia thus has implication for the development of dangerous at-risk behaviors, in particular substance use (and abuse) in the adolescent population. This has been established by large population-based studies [15,59,91,102]. Studies have suggested that the association between insomnia, risk-taking, and substance abuse during adolescence may be explained by the impact of insomnia on cognitive abilities and the reward system, resulting in a greater likelihood of engaging in risky behaviors. Interestingly, authors who investigated cognitive control and risk-taking in 14–16 year-old adolescents using functional magnetic resonance imaging (fMRI) found an association between poor sleep and greater risk-taking, perhaps due to an imbalance between mood and control systems worsened by sleep disruption [60]. Therefore, adolescents with poor sleep may exhibit increased arousal towards reward and impaired cognitive control and decision making, which results in a greater orientation towards risk and poorer decision-making abilities [102].

### 5.4. Suicide Risk

Suicide is the second/third leading cause of death among adolescents (and, surprisingly also among people aged over 10 years). Sleep disturbances remain independent predictors of suicidality in adolescents, even when analyses are adjusted for mood disorders. In a survey on a large sample of 10,123 adolescents, insomnia symptoms were significantly related to suicidality (suicidal ideation, suicidal plans, and suicidal attempts). The presence of insomnia symptoms was associated with a 6.2-fold increased risk of suicidal ideation, a 10.4-fold increased risk of making a suicide plan, and a 10.5-fold risk of making a suicide attempt, when compared to the absence of insomnia symptoms [85,103]. Goldstein and colleagues observed on a sample of autopsies of adolescents that sleep disturbances (insomnia and hypersomnia) were more frequent among the suicide victims than the controls, even when the sample was adjusted for the presence of affective disorders [104]. The data clearly show that the presence of insomnia substantially elevates the risk of these phenomena. Recently, it has been suggested that suicide prevention programs should include discussion and evaluation of sleep-related issues, including management of insomnia [105]. It is possible that this insomnia effect on suicide risk might be mediated by the effects of insomnia on arousal, emotional, and reward systems and decision-making processes.

## 6. The Effect of Insomnia Treatment on Mental Health in Adolescence

The literature reviewed so far highlights that sleep disturbances are likely a causal factor contributing to the onset of most mental health problems in adolescents (Figure 1), thus prompting the incorporation of routine assessment and treatment of sleep problems into the care pathway. Initiatives in targeting interventions on critical contributory causal factors such as insomnia and mental health problems have therefore been implemented. There is agreement in the literature that early intervention on sleep problems could be a preventive strategy for the onset of clinical disorders [106]. Notably, recent studies have shown a positive effect of insomnia treatment on mental health. However, most studies on this topic include only adults, and information on the treatment’s impact on the developmental trajectories of mental illness in the adolescent population is still scarce. The standardized guidelines for the diagnosis and treatment of insomnia recommend cognitive behavioral therapy for insomnia (CBT-I) as the first-line treatment for chronic insomnia in adults of any age. A pharmacological intervention (benzodiazepines, benzodiazepine receptor agonists, and some antidepressants) can be offered in the short-term treatment of insomnia (≤4 weeks) if cognitive behavioral therapy for insomnia is not sufficiently effective or not available [107].

Gebara and colleagues conducted a meta-analysis examining the effects of insomnia treatment on depression, indicating an improvement with moderate to large effect sizes (ESs) in depression, as measured by the Hamilton Depression Rating Scale and the Beck Depression Inventory, thus confirming that insomnia treatment has a positive effect on mood [108]. However, the interventions included both CBT and pharmacotherapy, and the populations varied widely (almost exclusively including individuals over 18 years); the effect of different treatments for insomnia were not compared nor was there clarification regarding the outcome in adolescents versus adults.

Regarding CBT-I, a study demonstrated lower rates of suicidal ideation at post-treatment and 1-year follow-up in adult patients with insomnia and suicidal ideation undergoing CBT-I [109]. Concerning adolescents, a study involving 116 subjects (mean age 15.9 years) specifically investigated whether CBT-I improves psychopathology and whether improvements can be attributed to reduced insomnia. The authors reported reductions not only in sleep problems but also in in psychopathological symptoms both in adolescents treated with face-to-face groups and with an at-distance CBT-I (which involves the use of the Internet) compared with controls at the stage of 2-month and 12-month follow-up [110].

Moreover, another study that aimed to evaluate the effects of psychological treatment on sleep in young people aged 15–22 who were at ultra-high risk of psychosis, reported significant improvements in sleep, mood, and functioning [111]. In the field of non-pharmacological approaches, there is emerging evidence that sleep problems can be treated using protocols that include a mindfulness component. Mindfulness meditation may be suitable for sleep problems as it aims to reduce hyperarousal and negative emotional states frequently reported by individuals with sleep problems [112,113,114]. Some studies proved that a multi-component group sleep intervention including cognitive-behavioral and mindfulness-based therapies could reduce sleep problems and anxiety symptoms [115] and demonstrated an improvement in behavior problems (social problems, attention problems, and aggressive behaviors) by enhancing perceived sleep quality in at-risk adolescents [114]. In addition, mindfulness-oriented meditation training has been indicated to be a promising tool for improving sleep quality and behavioral manifestations in ADHD [116].

Regarding pharmacological treatment, there are no randomized-controlled trials conducted in the adolescent population, but a few data are available in adults. A double-blind placebo-controlled parallel-group randomized-controlled trial evaluated whether treatment with controlled-release zolpidem in adults (18–65 years of age) with major depression, insomnia, and suicidal ideation could significantly reduce suicidal ideation when compared with use of a placebo. The study did not demonstrate a significant treatment effect on the Scale for Suicide Ideation, but the score reduction positively correlated with improved insomnia, with more significant benefits in patients with more severe insomnia. Thus, the authors concluded that co-prescribing a hypnotic while starting an antidepressant may benefit suicidal outpatients, especially those with severe insomnia [105].

In addition to insomnia, circadian rhythm disorders may be a transdiagnostic condition for many psychiatric disturbances in children and adolescents. These disorders may warrant a systematic evaluation in routine clinical practice and could be the target for specific treatments. The differential diagnosis between circadian rhythm disorders and insomnia is important for correct therapeutic intervention. Treatment of circadian rhythm disorders in adolescents involves a combination of chronotherapeutic strategies, including bright light therapy (evening light exposure delays the clock while morning light phase advances it), treatment with melatonin as a chronoregulator, and interventions on non-photic and extrinsic factors. These include establishing regular sleep patterns, limiting the use of technology (television, computers, electronic tablet devices) in the bedroom particularly during the hour before the desired sleep time, and avoiding caffeine, energy-dense foods and exercise near to the desired sleep time [100,117]. Interestingly, bright light therapy with a gradual advance protocol proved to be an effective adjunctive treatment resulting in a higher rate of remission of depression in adult patients with unipolar depression and evening chronotype [118]. Future studies should assess the impact of the treatment of circadian rhythm disorders on psychological distress and psychiatric disturbances in adolescents.

## 7. The Impact of COVID-19 Outbreak on Sleep and Mental Health in Adolescence

Regarding social changes and their effect on reshaping health, mental health, and the adolescent brain [24], the COVID-19 pandemic, along with the measures adopted to prevent its spread, has deeply impacted the quality of life and well-being of adolescents worldwide and so, therefore, can have a potential role in reframing not only social perspective but also neurobiological pathways in modern adolescents. The pandemic induced deep changes in lifestyle and daily habits of adolescents through prolonged school closure, changes in school routines, distance learning, social isolation, and increased interactions with parents and siblings. Moreover, adolescents reported reduced physical activity, and increased time spent being sedentary in front of a screen during the lockdown period [119,120,121,122]. Theoretically, such a condition, associated with the threat represented by the pandemic itself, may represent a significant risk factor for physical and mental health in adolescents. Consistently, adverse mental health symptoms have been frequently reported in adolescents during the pandemic, mainly in terms of depressive (and therefore suicidal) behaviors, self-harm, anxiety, and post-traumatic stress symptoms [123,124,125,126,127,128]. Around the world, high levels of uncertainty and psychological distress in adolescents during the pandemic have been consistently reported [129,130,131].

Sleep habits and problems in adolescence during the pandemic and their relationship with mental health have been widely investigated, mainly through self-reported measures, and a first line of evidence highlights a greater risk of sleep problems in adolescents associated with the COVID-19 pandemic [125,132,133,134,135,136]. In their meta-analysis on sleep disturbances during the pandemic in the general population, Jahrami and colleagues found that children and adolescents represented the second most affected group after COVID-19 patients, with an overall prevalence of sleep problems of approximately 46% [132]. From another meta-analytic work, the pooled prevalence of sleep disorders among children and adolescents was 42%, with age, educational levels, and female gender correlating with depressive symptoms (pooled prevalence of 31%), which had an increase of prevalence over time [137]. Several studies from different countries reported a high prevalence of sleep disorders and poor sleep quality in adolescents during the pandemic [133,134,135,136]. In a large USA sample, adolescents were more likely to exhibit moderate to severe symptoms of depression, anxiety, post-traumatic stress disorder, suicidal ideation or behavior, and sleep problems compared to adults [138]. Moreover, comparing two at-risk groups for sleep problems (i.e., late adolescents and the elderly) during the lockdown in Italy, Amicucci and colleagues found that adolescents exhibited more severe insomnia symptoms, worse subjective sleep quality, longer sleep latency, higher daytime dysfunction, and a more prevalent disruption of sleep habits, together with higher levels of depression and perceived stress; older adults showed shorter sleep duration, lower habitual sleep efficiency, and greater use of sleep medications. There was a stronger negative impact on sleep and mental health in adolescents than in the elderly [139]. Consistently, a high prevalence of clinical insomnia in Italian adolescents has been reported in different pandemic phases [140]. However, findings about the comparison between pre-pandemic and pandemic sleep quality seem conflicting [120,141].

It is worth noting that a strong relationship between mental health and sleep disturbances has been found in adolescence during the pandemic. In a large sample of Chinese adolescents and young adults, depression and anxiety were risk factors for insomnia symptoms [134]. Bacaro and coworkers found in Italian adolescents a strong correlation between insomnia symptoms and poor sleep hygiene, psychological distress, and emotional suppression [140]. In a large Korean sample, the likelihood of mental health problems (depressive symptoms, anxiety, suicidal ideation) was greater in adolescents with lower sleep duration and satisfaction compared to other age groups [142]. During the first lockdown in the UK, a strong relationship between sleep quality and perceived changes in happiness has been observed in school-aged children and adolescents [133]. Crucially, using daily reports for one week, Palmer and coworkers (2022) found in US adolescents during the pandemic that higher sleep disturbances predicted next-day socioemotional difficulties, which in turn predicted more sleep problems the following night, suggesting a bidirectional cyclic association between sleep difficulties and daytime emotional experience [143]. Interestingly, in a cross-sectional and prospective Chinese study on a group aged between 16 and 25 years old, sleep disturbances during the COVID-19 outbreak were associated with an increased risk of post-traumatic stress disorder and depression [144]. A three-wave longitudinal study (including the period before and during the pandemic) found an association between poor sleep and anxiety across the three waves in Chinese adolescents [145]. Using actigraphy and sleep logs before and during the COVID-19 pandemic in Canada, Gruber and coworkers found that pre-COVID-19 sleep duration and arousal at bedtime were predictors of adolescents’ perceived stress during the pandemic; poorer sleep quality and higher levels of arousal at bedtime were associated with greater perceived stress during the pandemic even when the authors controlled pre-pandemic emotional and behavioral issues, sleep duration and quality [146].

Interestingly, sleep problems in adolescents during the pandemic have also been frequently associated with specific daily habits. Specifically, longer screen time [135,136,147,148,149,150,151,152,153] and lower physical activity [149,154] have been associated with different measures of poor sleep quality. Considering the influence of these variables on sleep habits in adolescents [14,155,156,157], these findings are consistent with the increased screen time and sedentary behaviors observed in youth during the pandemic [119,120,121,122,141].

Beyond this type of detrimental effect of the pandemic on sleep quality associated with daily habits and psychological distress, a large number of studies conducted in several countries using different methodologies pointed to a beneficial effect of the lockdown period on adolescents’ sleep, reporting an increase of sleep duration associated with delayed bedtime and waketime [120,133,146,153,158,159,160,161,162,163,164,165,166]. Delayed sleep times and longer sleep duration in adolescents during the pandemic compared to the pre-pandemic period have also been observed objectively by means of actigraphy monitoring [146,167,168]. Moreover, reduced social jet lag [146,158,161,162,169] and diurnal sleepiness [141,160,162,169,170] have been observed. Such phenomena have been attributed to the delayed school start time during the lockdown period. Indeed, early school start time can contribute to the frequently observed chronic sleep deprivation in adolescents, conflicting with their biological rhythm [56,171,172]. On the other hand, later school start time has been associated with longer sleep duration, with a beneficial effect on health and academic performance [53,171,173,174]. In this view, adolescents may have benefitted from school closure during the COVID-19 pandemic, which may have favored a shift of the sleep schedule towards their own circadian preferences allowing longer sleep. It is worth noting that the longer sleep duration in adolescents during the COVID-19 lockdown, observed in comparison with a pre-lockdown group, was associated with better health-related quality of life and less caffeine consumption, while depressive symptomatology was negatively correlated with the same variables [163]. Such findings suggests that the positive effect on sleep duration, attributed to later wake times due to homeschooling, coexisted with a negative effect of psychological distress, likely associated with the adverse condition represented by the pandemic. Moreover, during post-lockdown periods, when schools were open again in a later phase of the pandemic, the beneficial effect of school closure on sleep disappeared, while adolescents exhibited chronic expressions of psychological distress and sleep problems [163,175].

The present literature suggests two opposite effects of the pandemic on adolescents’ sleep, exhibiting different relationships with mental health. On the one hand, there was a detrimental effect represented by the increased risk of sleep problems, influenced by changes in diurnal habits (i.e., screen time and physical activity) and bidirectionally related to psychological distress, substantially attributed to the adverse consequences of the pandemic. On the other hand, there was a beneficial effect represented by increased sleep duration and a delayed sleep timing, more fitting with adolescents’ circadian preferences and associated with better health-related quality of life, specifically attributed to the homeschooling period.

Along with the pandemic effects on healthy adolescents, the long-term consequences of the COVID-19 infection should be considered. The current literature does not report a unanimous definition of the Long-Covid Syndrome. However, the World Health Organization defines the long-COVID as “Post COVID-19 condition occurring in individuals with a history of probable or confirmed SARS-CoV-2 infection, usually 3 months from the onset of COVID-19 with symptoms and that last for at least 2 months and cannot be explained by an alternative diagnosis” [176]. Recent findings highlight that sleep alterations are one of the most common post-acute symptoms [177], and psychological factors such as stress and anxiety appear to be relevant predictors of sleep problems [178]. Although children and adolescents rarely develop a severe COVID-19 disease [179,180], adolescents are more frequently hospitalized than younger children [181]. Many findings show chronic consequences after the infection in this population [182,183], even when the acute phase of the disease is asymptomatic or mild, and a recent meta-analysis reveals that the prevalence of long-COVID among children and adolescents is around 25% [183]. Sleep problems, along with mood alterations and fatigue, are the most prevalent persistent symptoms [183]. Among sleep disorders, insomnia (accounting for 18.6% of adolescents) and poor sleep quality were found [167,168,169,170]. Not surprisingly, some authors hypothesized that the persistence of poor sleep quality might be driven by psychological processes rather than pathophysiological mechanisms [184]. While data is scarce, some findings highlight that insomnia in long-COVID adolescents is positively correlated with psychological measures (i.e., mental distress, anger, sadness, tension) and negatively correlated with happiness [185]. The current literature on this topic is still sparse, and no study has been carried out aiming at systematically investigating long-term sleep symptoms and their relationship with mental health status in adolescence after the COVID-19 infection, with no discrimination between children and adolescents (e.g., [186,187]). Moreover, the evaluation of the long-COVID symptoms has been conducted considering different time intervals from the acute infection.

## 8. Conclusions and Future Research

This narrative review aimed to summarize the knowledge in the field of adolescents’ mental health, in terms of the sleep changes that occur during this period, the impact of sleep deprivation and sleep disorders on mental health, and thus the risk of psychiatric disorders during this complex developmental period.

The stage of adolescence has survived among species and over many centuries to promote the survival of humankind, despite the high individual cost [188]. The interrelationships existing between the modern social context and the biological changes underpinning the acquisition of adults’ behavior are still poorly understood [24].

Although it is a healthy stage of life, adolescence is also a significantly vulnerable period, especially for at-risk individuals who face a poor affective or social context [29]. Sleep deprivation is partially a physiological step for this age group, but when enhanced by individual or social conditions it can be detrimental [7], and should be monitored and treated if severe or if associated with an insomnia disorder [15]. Studies addressing the macrostructure of adolescents’ sleep and, in particular, the role of REM sleep (as also a potential target of intervention) are needed.

The role of environment in shaping this disorder has been established and studies on targeted interventions are needed [27,36]. In this sense, the recent pandemic has created a heavy legacy of pathology that covers not only the Long-COVID condition but also more general impacts of the pandemic. Further investigations are needed in adolescents affected by the long-term consequences of COVID-19 to correctly identify sleep problems and manage them through appropriate intervention protocols. More attention should also be given to the consequences of sleep problems on long term psychological well-being of this age group. The consequences of not addressing adolescent well-being as a whole increase the risk of self-injury/suicidal behavior and/or extend the problem to adulthood, impacting both physical and mental health dramatically. Studies are also needed on adolescents to evaluate the efficacy of treatment of sleep disorders and programs of sleep hygiene through social/school policies [189]. Age-appropriate national policy interventions that focus on adolescents’ well-being and information are needed, and emotional education and sleep health programs should be implemented.

## Figures and Tables

**Figure 1 brainsci-13-00569-f001:**
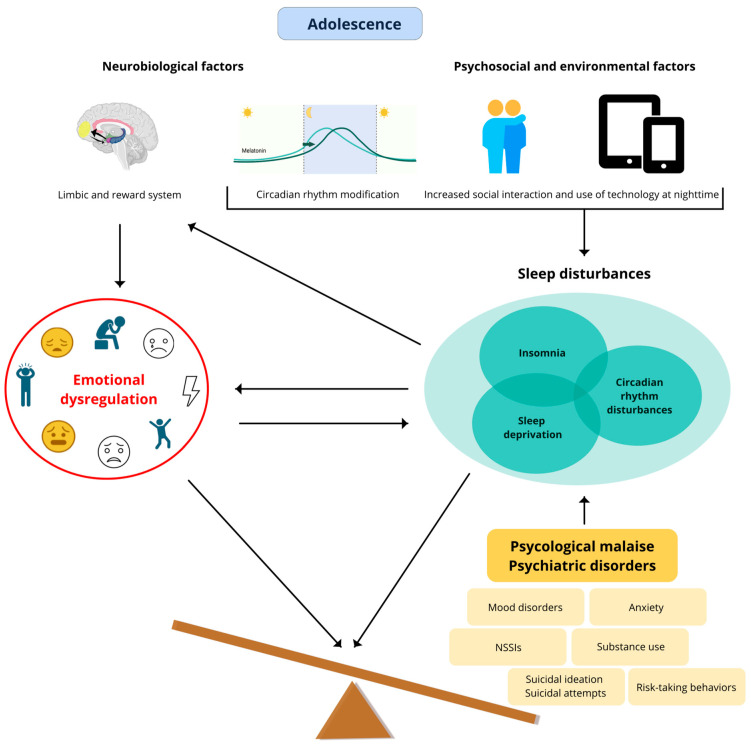
Neurobiological modifications in adolescents, circadian rhythm changes and mechanisms of development of psychiatric disorders. Legend. The adolescent brain faces many changes during a short period related to neurobiological modifications combined with psychosocial and environmental factors. Deep brain areas such as the limbic and reward system areas (ventral striatum and amygdala) increase their neural activity (top on the left). This temporary hyperfunction of dopaminergic circuits can lead to emotional dysregulation (center on the left), augmenting the risk of developing psychological malaise and psychiatric disorders (bottom on the right) [12]. Moreover, circadian rhythm modifications with delayed melatonin secretion (top center) occur during adolescence. These changes, together with environmental and psychosocial factors (increased social interaction and use of technology at nighttime) (top on the right), may result in sleep disturbances (sleep deprivation, circadian rhythm disorders, insomnia) (middle on the right) [13,14]. Of note, sleep loss may decrease functional connectivity between the top-down control regions of the prefrontal cortex and the amygdala [1]. Sleep disorders can drive/worsen emotional dysregulation, which in turn may contribute to the onset and/or maintenance of sleep disturbances. Moreover, sleep disorders may increase the risk of psychological problems and psychiatric disorders, which in turn may induce and/or maintain sleep disturbances [15,16,17].

## Data Availability

Not applicable.

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
