# Peer review of "Sleep Deprivation and Insomnia in Adolescence: Implications for Mental Health"

_brainsci, 2023, doi:10.3390/brainsci13040569_

Round 1

Reviewer 1 Report

Please see the attachment below

Author Response

Thank you for the opportunity to review this interesting narrative review on sleep issues and mental health in adolescence.

The topic is highly relevant, and the authors correctly pointed to largely unexplored areas of research that must be addressed in the near future. Briefly, they highlight the high prevalence of sleep problems during adolescence and the negative impact of sleep deprivation and insomnia on mental health. They conclude their review by proposing targeted sleep interventions on sleep during this vulnerable period of life, making this article particularly suitable for the Special Issue "Effects of Sleep Deprivation on Cognition, Emotion, and Behavior".

Dear reviewer, thank you for your comment.

This manuscript is well written and linearly organized. There are only a few minor issues, on my side:

  • Although the bibliography includes several up-to-date references, some statements are not followed by the proper citation – e.g., line 126-128, line 379-381; line 391-392/392/394

Dear reviewer, thank you for this observation. The statements were correctly followed by the right citation in the subsequent lines in the mentioned examples, except for line 379-381 that missed the citation of referral. The missing citation (n.105) has been implemented in the manuscript and highlighted in yellow both in the text body (line 419-421) and in the Reference section.

  • Conclusions should be improved to highlight more clearly the relevance of the review, summarize weaknesses/gaps in our current knowledge and more specific possible future lines of research.

Dear reviewer, thank you for your comment. We modified the Conclusion section according to your suggestions.

  • Minor typos should be corrected (e.g., which is usually preceded by a comma; line 27: ‘resume’: summarize? line 111: ‘select(ed?)’; line 178: ‘later’; line 190: risky should be placed after driving and sexual; line 312: extra-space after ‘of’; line 331: ‘of sleep’> ‘between sleep; line 379: ‘occurring’; line 414: ‘in’; 422: ‘controlled’: repeated twice?

Dear reviewer, thank you for your comment, we have extensively revised the English language of our manuscript and corrected the typos you have mentioned. You will find the misspelling and typos corrections highlighted in yellow throughout the manuscript.

  • Line 133: The adolescence time window has dilated during last decades due to the procrastination of adult life by enlarging the periods of economic and individual dependence from the original parental nucleus, thus potentially expanding also the period of major vulnerability [14,15]: for the sake of curiosity, which is the biological explanation of this phenomenon?

 Dear reviewer, thank you for your comment. As resumed by Crone and Dahl on at Rev Neurosc. in 2012,the relationships between these changes in the length, timing, nature and goals of adolescence and the brain changes associated with adolescence are not yet understood”. Developmental neuroimaging studies have shown a more complex model of brain maturation in adolescence that does not involve only a top-down model of cortical maturation of the decision-making dorsal prefrontal cortex. Other deep circuits involving ventral prefrontal cortex and limbic areas have different timing for maturing that occur at different stages of adolescence, that mediated by social and affective processing, through cascades of sexual hormones. Understanding these mechanisms happening in adolescence (a period of social-affective engagement and goal flexibility, but also of high health risk) implies therefore not only biological research but also a social-anthropological perspective in the sense that environment can shape brain modulation in a sort of bottom-up mechanism and therefore be a key for therapeutical approaches.  (Crone EA and Dahl RE.  Understanding adolescence as a period of social-affective engagement and goal flexibility. Nat Rev Neurosci. 2012 Sep;13(9):636-50.  doi: 10.1038/nrn3313). We added a paragraph expanding this concept at lines 138-156.

  • Section 5.1. Insomnia in adolescents may increase the risk of psychiatric disorders: Here, I suggest to highlight more the bidirectionality of the connection and discuss more about increasing the risk versus trigger/anticipate the disorder in predisposed subjects. Furthermore, it might be interesting to shortly discuss the possible differential role of REM versus REM sleep deprivation (here or somewhere else in the manuscript).

Dear reviewer, thank you for the suggestion. We discussed the mutual interaction between sleep disorders and psychiatric disorders by also highlighting it in Figure 1 and the legend. In addition, we discussed the specific role of REM sleep in the text (please see lines 162-170; lines 243-247).

  • Line 590: Sleep deprivation is partially a physiological step for this age, but when is enhanced by 588 individual or social conditions it can be detrimental [7], and should be monitored and 589 treated if it configures as an insomnia disorder [47] > I suggest to rather rephase as ‘if severe or if associated with an insomnia disorder’ to avoid confusion.

Dear reviewer, thank you for your suggestion, we have modified the sentence according to your proposal of rephrasing. Find the novel version of the sentence highlighted in yellow at lines 641-644.

  • The COVID section is interesting but rather long, it might be slightly shortened up.

Dear reviewer, thank you for your comment, although literature on the topic is very rich and very uneven, and although it has been suggested from other reviewers to enrich the bibliography on the topic, as far as it concerned to us, we shorten a bit this section. Find modifications throughout the section barred and highlighted in yellow.

  • The figure is not embedded in the text and I could not revise it.

Dear reviewer, thank you for your comment. We have updated the revised manuscript with the figure and the final version of its legend.

Reviewer 2 Report

No extra recommendation

Author Response

No extra recommendation

Thank you for having revised our manuscript. We have extensively revised the English language and style, correcting fine/minor spell checks, Moreover, we modified the previous version of the manuscript to improve its contribution and scientific relevance in the field, its organization, and we also updated the reference list according to these modifications.

Reviewer 3 Report

Review Process for Brain Sciences

Title: Sleep deprivation and insomnia in adolescence: implication for mental health

Authors: Uccella, Cordani, et al.

Manuscript ID: brainsci-2219074

The manuscript is a review aiming to present how the brain physiology during adolescent window has an effect on sleep changes and, in turn, the impact of sleep deprivation on mental health.

The paper dealt with an important topic in the field with relevant implication at individual and societal level. However, I think that this manuscript is a (good) book chapter but not a real review. Although someone can think that this paper is “close to” a narrative review, all features of a narrative review are missing (search criteria, database used, keywords used, etc.). Furthermore, the authors in several paragraphs correctly quoted important reviews and meta-analysis but for the reader the real contribution of this paper is lacking. Related to this point, different paragraphs are not linked together, making confusion for the reader (e.g., the last paragraph about COVID-19 effect on adolescent sleep and mental health after paragraph 6 and far away from paragraph 5). Thus, I suggest to define what is useful to review (e.g., the effect of physiological brain changes on sleep quantity and quality during adolescence), and, for example, to appraise the topic using the Consensus-Based Standards for the Selection of Health Status Measurement Instruments (COSMIN) checklist.

Additional concerns:

Page 2, lines 90-94, I am not so sure that the typical adolescent behavior is uniquely related to physiological changes during adolescence. An environmental aspect is relevant. Thus, here (and in other parts of the manuscript), I suggest to use more caution in affirming several statements. For example, the increase of social interactions with peers is not only related to promote sexual interaction.

Page 8, paragraph 6: Beyond pharmacological and CBT-I treatment of insomnia, in the last (about) 10 years additional non-pharmacological approaches are advanced and used in adolescent sample. Probably, a consideration of these additional insomnia treatment could render this paragraph to be complete.

Although the impressive reference section, I think that several (too many) papers related to chronobiological and chronopsychological changes during adolescence are missing.

Author Response

The manuscript is a review aiming to present how the brain physiology during adolescent window has an effect on sleep changes and, in turn, the impact of sleep deprivation on mental health.

Dear reviewer, thank you for having revised our manuscript.

The paper dealt with an important topic in the field with relevant implication at individual and societal level. However, I think that this manuscript is a (good) book chapter but not a real review. Although someone can think that this paper is “close to” a narrative review, all features of a narrative review are missing (search criteria, database used, keywords used, etc.).

Dear reviewer, thank you for your comment. Indeed, we did not aim to write a systematic review or a scoping of the literature (thus requiring the following of precise guidelines such as the ones proposed by the PRISMA guidelines - Equator network, the Chocraine methods, or other useful checklists for these types of articles.

Narrative reviews, according to the Medical Subject Headings (MeSH) scope notes (National Center for Biotechnology Information. Review Literature as Topic. https://www.ncbi.nlm.nih.gov/mesh/?term=literature%20review%20as%20a%20topic accessed the 16th of March, 2023) “provide an examination of recent or current literature. These articles can cover a wide range of subject matter at various levels of completeness and comprehensiveness based on analyses of literature that may include research findings. The review may reflect the state of the art and may also include reviews as a literary form”.

Indeed, although lacking an explicit intent to maximize scope or to analyse data collected, narrative reviews have the advantage of allowing for consolidation, for building on previous work, for summation, for avoiding duplication and for identifying omissions or gaps ( Grant et al: A Typography of Reviews published in Health Information and Libraries Journal, 2009, 26:2).

Therefore, as also not required from the Journal (Brain Sciences) policies for authors (“[..]Reviews offer a comprehensive analysis of the existing literature within a field of study, identifying current gaps or problems. They should be critical and constructive and provide recommendations for future research. No new, unpublished data should be presented. The structure can include an Abstract, Keywords, Introduction, Relevant Sections, Discussion, Conclusions, and Future Directions, with a suggested minimum word count of 4000 words [..]”; see https://www.mdpi.com/about/article_types - MDPI Instruction for Authors) we decided to leave the structure of the manuscript as it was, and to expand the Conclusions and future research section, to better highlight the relevance of our review and to  summarize weaknesses/gaps in our current knowledge and more specific possible future lines of research. The modifications of this section are highlighted in yellow.

Furthermore, the authors in several paragraphs correctly quoted important reviews and meta-analysis but for the reader the real contribution of this paper is lacking.

As above mentioned, we decided to expand the Conclusions and future research section, to better highlight the relevance of our review and to summarize weaknesses/gaps in our current knowledge and more specific possible future lines of research. The modifications of this section are highlighted in yellow.

Related to this point, different paragraphs are not linked together, making confusion for the reader (e.g., the last paragraph about COVID-19 effect on adolescent sleep and mental health after paragraph 6 and far away from paragraph 5).

Dear reviewer, thank you for your comment. Indeed, we tried to homogenize the contents as much as possible. We tried our best for make the work as linear as possible. In this revised version we better explicated the transitions from a paragraph to another and we better summarize the main results in the Conclusions section. Find changes throughout the manuscript highlighted in yellow.

Thus, I suggest to define what is useful to review (e.g., the effect of physiological brain changes on sleep quantity and quality during adolescence), and, for example, to appraise the topic using the Consensus-Based Standards for the Selection of Health Status Measurement Instruments (COSMIN) checklist.

Dear reviewer, thank you for your comment. The Consensus-Based Standards for the Selection of Health Status Measurament Instruments (COSMIN) checklist is a method developed starting from the paradigm of the Delphi study approach, a structured communication technique or method, originally developed as a systematic, interactive, and forecasting. This was not the aim of our review that was built, as mentioned in the Abstracts, to summarize the knowledge and the gaps in the field of adolescents’ sleep deprivation and its possible consequences on their mental health.

Additional concerns:

Page 2, lines 90-94, I am not so sure that the typical adolescent behavior is uniquely related to physiological changes during adolescence. An environmental aspect is relevant. Thus, here (and in other parts of the manuscript), I suggest to use more caution in affirming several statements. For example, the increase of social interactions with peers is not only related to promote sexual interaction.

Dear reviewer, thank you for your comment. In lines 90-94 human adolescence is put in parallel with the ones of other mammals. Considerations on social environment and social perspective are therefore discussed throughout the manuscript. In the conclusions we further explicated these concepts, with attention to prudent statements. Find the new Conclusion section highlighted in yellow at the end of the manuscript.

Page 8, paragraph 6: Beyond pharmacological and CBT-I treatment of insomnia, in the last (about) 10 years additional non-pharmacological approaches are advanced and used in adolescent sample. Probably, a consideration of these additional insomnia treatment could render this paragraph to be complete.

Dear reviewer, thank you for your comment which allowed us to modify the chapter and add useful information on the most important non-pharmacological approaches for treating insomnia. Please see the changes highlighted in yellow in chapter 6 (459-469 lines and 483-495 lines).

Although the impressive reference section, I think that several (too many) papers related to chronobiological and chronopsychological changes during adolescence are missing.

Dear reviewer, thank you for your comment. We have made changes to chapter 3 and added some references as suggested at lines 171-184.

Reviewer 4 Report

Sleep deprivation and insomnia in adolescence: implications for mental healthbrainsci-2219074

This review aimed to summarize the state of art of the topic about the effects of sleep deprivation and insomnia on mental health in adolescence. Overall, this topic is interesting and important and this manuscript is comprehensive and quite updated, especially during COVID-19. However, some minor concerns appeared after reading the whole manuscript.

1. The literature and data about mental health in adolescence during COVID-19 need to be updated.

Deng, J., Zhou, F., Hou, W., Heybati, K., Lohit, S., Abbas, U., ... & Heybati, S. (2022). Prevalence of mental health symptoms in children and adolescents during the COVID‐19 pandemic: A meta‐analysis. Annals of the New York Academy of Sciences.

Bersia, M., Koumantakis, E., Berchialla, P., Charrier, L., Ricotti, A., Grimaldi, P., ... & Comoretto, R. I. (2022). Suicide spectrum among young people during the COVID-19 pandemic: A systematic review and meta-analysis. EClinicalMedicine54, 101705.

Wan Mohd Yunus, W. M. A., Kauhanen, L., Sourander, A., Brown, J. S., Peltonen, K., Mishina, K., ... & Gyllenberg, D. (2022). Registered psychiatric service use, self-harm and suicides of children and young people aged 0–24 before and during the COVID-19 pandemic: a systematic review. Child and adolescent psychiatry and mental health16(1), 1-13.

2. Some related references need to be reviewed and discussed, such as,

Tarokh, L., Saletin, J. M., & Carskadon, M. A. (2016). Sleep in adolescence: Physiology, cognition and mental health. Neuroscience & Biobehavioral Reviews70, 182-188.

Jamieson, D., Broadhouse, K. M., Lagopoulos, J., & Hermens, D. F. (2020). Investigating the links between adolescent sleep deprivation, fronto-limbic connectivity and the onset of mental disorders: a review of the literature. Sleep Medicine66, 61-67.

Galván, A. (2020). The need for sleep in the adolescent brain. Trends in cognitive sciences24(1), 79-89.

Hayes, B., & Bainton, J. (2020). The impact of reduced sleep on school related outcomes for typically developing children aged 11–19: A systematic review. School Psychology International41(6), 569-594.

3. Where is Figure 1

Author Response

This review aimed to summarize the state of art of the topic about the effects of sleep deprivation and insomnia on mental health in adolescence. Overall, this topic is interesting and important and this manuscript is comprehensive and quite updated, especially during COVID-19.

Dear Author, thank you for having revised our manuscript.

However, some minor concerns appeared after reading the whole manuscript.

  1. The literature and data about mental health in adolescence during COVID-19 need to be updated.

Deng, J., Zhou, F., Hou, W., Heybati, K., Lohit, S., Abbas, U., ... & Heybati, S. (2022). Prevalence of mental health symptoms in children and adolescents during the COVID‐19 pandemic: A meta‐analysis. Annals of the New York Academy of Sciences.

Dear reviewer, thank you for your interesting suggestion, we have updated our manuscript with the information coming from this article. Find the citation at lines 519-523

Bersia, M., Koumantakis, E., Berchialla, P., Charrier, L., Ricotti, A., Grimaldi, P., ... & Comoretto, R. I. (2022). Suicide spectrum among young people during the COVID-19 pandemic: A systematic review and meta-analysis. EClinicalMedicine, 54, 101705.

Dear reviewer, thank you for your interesting suggestion, we have updated our manuscript with the information coming from this article. Find the citation at lines 510-511

Wan Mohd Yunus, W. M. A., Kauhanen, L., Sourander, A., Brown, J. S., Peltonen, K., Mishina, K., ... & Gyllenberg, D. (2022). Registered psychiatric service use, self-harm and suicides of children and young people aged 0–24 before and during the COVID-19 pandemic: a systematic review. Child and adolescent psychiatry and mental health, 16(1), 1-13.

Dear reviewer, thank you for your interesting suggestion, we have updated our manuscript with the information coming from this article. Find the citation at lines 510-511

  1. Some related references need to be reviewed and discussed, such as,

Tarokh, L., Saletin, J. M., & Carskadon, M. A. (2016). Sleep in adolescence: Physiology, cognition and mental health. Neuroscience & Biobehavioral Reviews, 70, 182-188.

Dear reviewer, thank you for your interesting suggestion, we have updated our manuscript with the information coming from this article. Find the citation along the second and the third paragraphs.

Jamieson, D., Broadhouse, K. M., Lagopoulos, J., & Hermens, D. F. (2020). Investigating the links between adolescent sleep deprivation, fronto-limbic connectivity and the onset of mental disorders: a review of the literature. Sleep Medicine, 66, 61-67.

Dear reviewer, thank you for your interesting suggestion, we have updated our manuscript with the information coming from this article. Find the citation along the third paragraphs.

Galván, A. (2020). The need for sleep in the adolescent brain. Trends in cognitive sciences, 24(1), 79-89.

Dear reviewer, thank you for your interesting suggestion, we have updated our manuscript with the information coming from this article. Find the citation along the third paragraphs.

Hayes, B., & Bainton, J. (2020). The impact of reduced sleep on school related outcomes for typically developing children aged 11–19: A systematic review. School Psychology International, 41(6), 569-594.

Dear reviewer, thank you for your interesting suggestion, we have updated our manuscript with the information coming from this article. Find the citation at line 273-274.

  1. Where is Figure 1?

Dear reviewer, thank you for your comment. We have updated the revised manuscript with the figure and the final version of its legend.

Reviewer 5 Report

This is a very good review about sleep disorders and their impact on mental health among adolescents. There are no major issues I see. However, I would like to mention one point. Authors use a very complex (very good) language with several words which an usual reasder whos english language is not native, may be have problems to understand. This is not an issue; however I wanted to let authirs know that many readers are in other countries of the world and of courise they understand English, but they will use a dictionary when read this article:)

Author Response

This is a very good review about sleep disorders and their impact on mental health among adolescents. There are no major issues I see. However, I would like to mention one point. Authors use a very complex (very good) language with several words which an usual reader whose English language is not native, may be have problems to understand. This is not an issue; however I wanted to let authors know that many readers are in other countries of the world and of course they understand English, but they will use a dictionary when read this article:)

Dear Author, thank you for having revised our manuscript and for your comment. We have extensively revised the English language of our manuscript to make it more readable for the readers.

Round 2

Reviewer 3 Report

The authors have improved the quality of the manuscript and for me it is suitable for the publication